# Characteristics of Nephroblastoma/Nephroblastomatosis in Children with a Clinically Reported Underlying Malformation or Cancer Predisposition Syndrome

**DOI:** 10.3390/cancers13195016

**Published:** 2021-10-07

**Authors:** Nils Welter, Angelo Wagner, Rhoikos Furtwängler, Patrick Melchior, Leo Kager, Christian Vokuhl, Jens-Peter Schenk, Clemens Magnus Meier, Stefan Siemer, Manfred Gessler, Norbert Graf

**Affiliations:** 1Department of Pediatric Oncology and Hematology, Saarland University, 66421 Homburg, Germany; nils.welter@uks.eu (N.W.); s9aowagn@stud.uni-saarland.de (A.W.); rhoikos.furtwaengler@uks.eu (R.F.); 2Department of Radiation Oncology, Saarland University, 66421 Homburg, Germany; patrick.melchior@uks.eu; 3St. Anna Kinderspital, Department of Pediatrics, Medical University Vienna, Kinderspitalgasse 6, 1090 Vienna, Austria; leo.kager@stanna.at; 4Section of Pediatric Pathology, University of Bonn, Venusberg-Campus 1, 53127 Bonn, Germany; Christian.vokuhl@ukbonn.de; 5Division of Pediatric Radiology, Clinic for Diagnostic and Interventional Radiology, University of Heidelberg, Im Neuenheimer Feld 430, 69120 Heidelberg, Germany; jens-peter.schenk@med.uni-heidelberg.de; 6Department of General, Visceral, Vascular and Pediatric Surgery, Saarland University, 66421 Homburg, Germany; clemens-magnus.meier@uks.eu; 7Department of Urology and Pediatric Urology, Saarland University, 66421 Homburg, Germany; stefan.siemer@uks.eu; 8Developmental Biochemistry and Comprehensive Cancer Center Mainfranken, Theodor-Boveri-Institute/Biocenter, University of Würzburg, 97074 Würzburg, Germany; gessler@biozentrum.uni-wuerzburg.de

**Keywords:** nephroblastoma, clinical malformations, cancer predisposition syndromes, tumor surveillance, outcome

## Abstract

**Simple Summary:**

It is well known that different cancer predisposition syndromes are associated with characteristic WT-features. The following findings from our retrospective analysis of patients with nephroblastoma treated according to the SIOP/GPOH trials between 1989 and 2017 are relevant: (1) The outcome of patients with a cancer predisposition syndrome is not always favorable despite early diagnosis, small tumors and less metastatic disease. This finding is partly depending on complications related to the underlying syndrome. (2) Predisposition syndromes seem to be underdiagnosed as several clinical and pathological features of Wilms tumor being clearly linked to a cancer predisposition syndrome did not lead to genetic counseling before and after WT diagnosis. As a conclusion, in children with a nephroblastoma and specific clinical and pathological features that are in line with a nephroblastoma cancer predisposition syndrome such a syndrome should always be considered and ruled out if unknown at the time of tumor diagnosis.

**Abstract:**

(1) Background: about 10% of Wilms Tumor (WT) patients have a malformation or cancer predisposition syndrome (CPS) with causative germline genetic or epigenetic variants. Knowledge on CPS is essential for genetic counselling. (2) Methods: this retrospective analysis focused on 2927 consecutive patients with WTs registered between 1989 and 2017 in the SIOP/GPOH studies. (3) Results: Genitourinary malformations (GU, *N* = 66, 2.3%), Beckwith-Wiedemann spectrum (BWS, *N* = 32, 1.1%), isolated hemihypertrophy (IHH, *N* = 29, 1.0%), Denys-Drash syndrome (DDS, *N* = 24, 0.8%) and WAGR syndrome (*N* = 20, 0.7%) were reported most frequently. Compared to others, these patients were younger at WT diagnosis (median age 24.5 months vs. 39.0 months), had smaller tumors (349.4 mL vs. 487.5 mL), less often metastasis (8.2% vs. 18%), but more often nephroblastomatosis (12.9% vs. 1.9%). WT with IHH was associated with blastemal WT and DDS with stromal subtype. Bilateral WTs were common in WAGR (30%), DDS (29%) and BWS (31%). Chemotherapy induced reduction in tumor volume was poor in DDS (0.4% increase) and favorable in BWS (86.9% reduction). The event-free survival (EFS) of patients with BWS was significantly (*p* = 0.002) worse than in others. (4) Conclusions: CPS should be considered in WTs with specific clinical features resulting in referral to a geneticist. Their outcome was not always favorable.

## 1. Introduction

Nephroblastoma or Wilms tumor (WT), the most common kidney tumor in childhood [1], can be cured in more than 90% today [2,3,4]. According to SIOP clinical studies and trials patients are diagnosed by imaging studies alone and preoperatively treated with AV (actinomycin and vincristine for 4 weeks) with localized or with (AV plus doxorubicin for 6 weeks) with metastatic tumors. During the registration process of patients, data on the kind of a cancer predisposition syndromes (CPS) or a malformation are provided by the treating hospital without further specifying malformations. In addition, participation in a surveillance protocol before the diagnosis in case of a CPS is registered in the database.

With 8 to 17% overall and up to 24% in bilateral WTs it has one of the highest association rates with congenital anomalies of all childhood cancers [5,6]. Such malformations and CPS related to the development of WTs are characterized by genetic or epigenetic alterations. For example, the WAGR syndrome, is clinically defined by a variable occurrence of WT in combination with aniridia, genitourinary malformations and a range of developmental delays [7,8,9]. It is caused by chromosome 11p13 deletions, including WT1 and neighboring genes, whereas Denys-Drash syndrome (DDS) is due to a dominant-negative WT1 mutation. DDS is characterized by the triad of WT, nephropathy and, if applicable, male pseudohermaphroditism [10,11,12]. In addition, genitourinary malformations (GU) have been linked to WT1 mutations [13,14]. Isolated hemihypertrophy (IHH) [15,16] and the Beckwith-Wiedemann spectrum (BWS) are overgrowth syndromes with elevated risk to develop WT. BWS shows a high variability of macroglossia, abdominal wall defects, visceromegaly, gigantism and hypoglycaemia caused by genetic and epigenetic alterations at 11p15.3 [12,17,18]. Other WT associated syndromes such as Perlman syndrome or Simpson-Golabi-Behmel syndrome are much rarer and have a different genetic background. In this paper we focus on the five most frequent WT malformations or CPS, namely WAGR, DDS, GU, IHH and BWS, to compare their clinical, pathological and outcome data with data from WTs without a known CPS.

## 2. Materials and Methods

We conducted a retrospective investigation on data of 2927 patients with WT and/or nephroblastomatosis from Germany, Austria and Switzerland enrolled in the SIOP/GPOH 9, 93-01 and 2001 studies between 1989 and 2017. Details of their treatment protocols have been reviewed previously [19]. Ethical approval was obtained from the Ärztekammer des Saarlandes (No: 136/01 from 20 September 2002 and, 248/13 from 13 January 2014). All parents or legal guardians of the affected children gave informed consent for study participation.

Pseudonymized data of all patients were stored in a central and encrypted SQL database. All patients identified in the database with a clinically documented malformation or CPS were reviewed by NW and NG, and details on these patients—including presentation, treatments and outcome—were collected from the SIOP-RTSG/GPOH database and, retrospectively, from status report forms, radiology, pathology and surgery reports, progress letters and telephone notes available at the data center. The identification of patients with malformations or CPS was based solely on clinical data provided by the registration CRF where associated congenital malformations or a syndrome were asked specifically for Aniridia, WAGR, genitourinary malformations, Denys Drash syndrome, BWS, IHH, Perlman syndrome. Free text could specify other malformations or syndromes that are not listed. This information is based on clinical characteristics. No information is provided if the syndrome was confirmed by genetic analysis. Patients with WAGR were also included in the paper by Hol et al. [8]. Tumor volume was calculated from imaging studies using the ellipsoid formula in those patients were CT or MRI of the tumor was available. Nephroblastomatosis was based on reference histology defined as multiple or diffuse nephrogenic rests but not further specified as perilobar, intralobar or both, as this information was not available for all patients with nephroblastomatosis. For statistical analysis all data were anonymized. IBM SPSS Statistics, version 25 and 27, was used for descriptive analyses (histograms, boxplots, pie charts, frequency charts and bar charts) and statistical comparisons (T-test for independent samples, Levene test, Chi-square test, multivariate analysis and Kaplan-Meier survival curves with Log Rank). *p*-values below 0.05 were considered as statistically significant. Overall survival (OS) included the time period between diagnosis and death of any reason, and event free survival (EFS) between diagnosis and any event, including recurrence of WT or nephroblastomatosis, death or loss to follow up.

## 3. Results

### 3.1. Characteristics of Study Population

An underlying malformation or syndrome was recorded in 198 out of all 2927 (6.8%) patients (Table 1). Bilateral disease occurred in 253 (8.6%) of patients and 29 patients with CPS or malformation were included in a surveillance program before diagnosis of a WT and/or nephroblastomatosis. In 137/2927 (4.6%) patients nephroblastomatosis was diagnosed, either isolated (73; 2.4%) or in conjunction with WT (64; 2.1%). This investigation highlights specifically a sub-cohort of 171 patients, who presented with one of the five most frequent malformations or syndromes (Table 1), that is GU (*N* = 66), BWS (*N* = 32), IHH (*N* = 29), DDS (*N* = 24) and WAGR syndrome (*N* = 20). In addition, 27 patients were diagnosed with a variety of other malformations or syndromes (Appendix A). Interestingly, there was no increase of the percentage of patients with CPS or GU over time. Up to year 2000, 54.9% of CPS or GU were diagnosed and 45.1% in the following years.

### 3.2. Ultrasound Surveillance Every 3 Months

Altogether 29 (14.6%) patients had been screened by ultrasound every 3 months after the diagnosis of a CPS. The highest screening frequency resulting in the diagnosis of WT/nephroblastomatosis was reported in patients with WAGR (40%) and BWS (31.3%). (Table 1). No data were available on why not all children with an underlying syndrome were included in a screening program.

### 3.3. Gender Distribution and Age at Diagnosis of WT/Nephroblastomatosis

Gender distribution in the whole cohort of patients with syndrome-associated WT is similar to the total group of patients with WT/nephroblastomatosis with a predominance of females (51.4% females vs. 48.1% males, 0.5% gender not known) with the exception of GU cases. 43 males (3.2% of 1353) and 23 females (1.5% of 1562) were affected by GU malformations. In DDS a slight male predominance has also been observed, but without statistical significance.

With a median age of 24.5 months (mean: 40.3 +/− 36.7 months), patients with associated malformations or syndromes were significantly (*t*-test: *p* < 0.001) younger at diagnosis of WT/nephroblastomatosis than patients without a malformation or syndrome (median age: 39.0 months; mean age: 50.0 +/− 51.6 months). Patients with WAGR (median age: 21 months; mean age: 23.8 +/− 9.0 months) and DDS (median age: 16.0 months; mean age: 16.7 +/− 12.2 months) were even significantly younger at diagnosis than patients with other syndromes (Figure 1).

### 3.4. Bilaterality

There was a statistically significant higher incidence of bilaterality in patients with (21.2%) than without a syndrome (7.4%) (*p* < 0.05), especially in patients with BWS (31.3%), WAGR (30.0%) and DDS (29.2%) compared to other patients with WT and/or nephroblastomatosis (Table 1).

### 3.5. Metastatic Disease in Patients with CPS or Malformations

In the whole group of patients 18% (529/2927) had a primary metastatic disease (stage IV) while in the cohort of patients with the five most common WT associated syndromes, primary metastatic disease was observed in less than 8.0% (14/171) (*p* < 0.001) with the exception of IHH (17.2%, *p* = 0.014) (see Appendix A).

### 3.6. Histology

According to the SIOP studies and trials histology of WT is classified in low, intermediate and high risk, depending on the availability of diffuse anaplasia, the percentage of necrosis and the percentage of blastema, epithelia and stroma in the vital tumor part after preoperative chemotherapy. Blastemal type WT and diffuse anaplasia are high risk tumors. Stromal type is mainly associated with WT1 mutations and is not responding on preoperative chemotherapy despite the fact that patients with stromal type WT have an excellent outcome in case of a localized tumor. The histological risk group together with the local and overall stage defines postoperative treatment. In addition, information about nephrogenic rests or nephroblastomatosis are provided. Mixed type, an intermediate risk tumor, is, with the exception of WAGR syndrome, the most common histological subtype for all WTs with or without syndromes. Patients with CPS are significantly more likely to have isolated nephroblastomatosis. In particular, a significantly increased proportion of isolated nephroblastomatosis is observed in WAGR, BWS and IHH (Table 2).

There was also a statistically significant association of IHH with the blastemal subtype after preoperative chemotherapy (*p* = 0.040) and of DDS with stromal subtype (*p* < 0.001) (Table 3).

### 3.7. Tumor Volume

Tumor volume (TV) at diagnosis and after preoperative chemotherapy was available in 1798 of 2927 (61.1%) patients (in 1698 patients without and in 91 with CPS or GU) (Table 4). In children with WT and CPS or GU TV at diagnosis was significantly lower than in patients without (349.4 mL vs. 487.5 mL; *p* < 0.001) (Table 4). Furthermore, with the exception of DDS a significant TV reduction can be achieved by preoperative chemotherapy in WTs with CPS or GU with the largest effect of 86.9% in patients with BWS showing an average TV after preoperative chemotherapy of only 38.3 mL (Table 4). In contrast, in DDS no real change of TV under preoperative chemotherapy was observed (Table 4).

Tumor volume at diagnosis in the 11 patients with CPS undergoing surveillance and treated with preoperative chemotherapy is significantly smaller (TV in CPS patients with surveillance: mean 62.7 mL, median: 21.3 mL. TV in CPS patients without surveillance: mean: 388.8 mL, median: 321.5 mL) (Figure 2).

### 3.8. Outcome

There was no statistically significant influence on EFS for the whole group of patients with a CPS (Figure 3A). However, patients with BWS showed a significantly worse EFS (Figure 3B) and a higher relapse rate (34.4%) compared to other patients with WT and/or nephroblastomatosis (13.7%). Out of 22 patients with BWS and only unilateral disease at diagnosis 5 patients relapsed of whom 3 showed metachronic disease (3, 4.5 and 6 years after initial diagnosis). One of these 3 patients developed also lung and liver metastasis and died 6 years after diagnosis. Of the other two relapsed patients one patient developed a local relapse in the same kidney and the other one developed lung metastasis without local or metachronic relapse and both survived. Apart from these three patients, metachronic disease occurred only in two further patients with CPS, one with WAGR syndrome and one with IHH. The contralateral kidney tumors were diagnosed in these patients 7 years (WAGR) and 10 months (IHH) after initial diagnosis, respectively. Further analysis suggested that EFS tends to be worse in patients with nephroblastomatosis and a syndrome than in patients without nephroblastomatosis (Figure 3C), particularly if they had developed WT in addition (Figure 3D). Appendix A gives an overview of outcome data. The only significant difference was seen in BWS for 5- and 10y EFS.

In a multivariate analysis of all patients, nephroblastomatosis and bilaterality had a significant influence on the risk of relapse and death. If only CPS and GU WT patients were considered, such risk was only found for relapse but not for death (Table 5).

## 4. Discussion

In our retrospective analysis we found that 5.8% of patients with WT and/or nephroblastomatosis are associated with the top five syndromes (WAGR, BWS, DDS, IHH and GU) in agreement with previous literature. With the exception of patients with GU and DDS, female patients are more frequently affected. Patients with syndromes show smaller TVs both at diagnosis and after preoperative chemotherapy, which might be due to the inclusion in a screening program [20]. The statistically significant lower frequency of metastatic disease at diagnosis in patients with a syndrome does not translates into a better EFS. Therefore, other factors such as nephroblastomatosis and comorbidities must be considered to explain their EFS, especially in patients with BWS. (Table 6).

### 4.1. Prevalence and Surveillance

The prevalence of syndromes in patients with WT is lower in our series compared to the 8–17% in the literature [5,6]. This may be due to an underreporting in our retrospective multicenter study where standardized reporting was carried out at diagnosis, hence early in life with a probably incompletely symptomatology. BWS, for example, a syndrome with variable features, is described with a prevalence of 1 up to 8% in other studies [12,18,29]. Therefore, not all patients with a WT CPS are included in ongoing ultrasound screening programs. In our data the screening rate is depending on the clinical symptomatology and highest in WAGR with 40% (Table 1). MacFarland et al. reported 12 patients diagnosed with BWS after a WT was already known [32]. This may explain our low prevalence of 1.1% and also undiagnosed BWS in other studies. Clinicians need to recognize subtle manifestations of syndromes in WT patients to not overlook them. For clinical diagnosis of BWS a new consensus statement has been published in this respect [33]. Knowledge about specific associations between different syndromes and WT will allow an earlier diagnosis of such WT, with CPSs demanding genetic testing, counselling and subsequently screening programs. In this respect, it is important to separate clearly between the WT1 associated CPS (DDS and WAGR) and the imprinting disorders (BWS and IHH) showing differences in WT characteristics, e.g., age at diagnosis or the response to preoperative chemotherapy as shown in our analysis.

### 4.2. Age at Diagnosis of WT/Nephroblastomatosis

Of all patients with syndromes, those with DDS are diagnosed the earliest followed by WAGR [9]. Patients with GU and BWS also tend to be diagnosed earlier than those without syndromes. The late median diagnosis at 45 months in IHH compared to other syndromes suggests that IHH manifests itself clinically rather subtly [12,30] which is why no surveillance for WT was carried out. Therefore, our results suggest that early age at diagnosis of a WT without a syndrome should always raise awareness of a CPS. In addition, in patients with the diagnosis of such a syndrome screening for WT needs to start early and regularly up to the age when the manifestation of a WT becomes more unlikely [20]. According to our data, for patients with DDS and WAGR such a screening may stop already at the age of 4 or 5 years as also recommended by Hol et al. [8], whereas in the other syndromic patients screening should continue at least up to the age of 7 years as also consented for BWS [20] (Figure 1). In the work of Diller et al. the average age of diagnosis of patients with GU was 13 months and thus significantly earlier compared to our data (42.9 months). This difference in median age may be related to a different approach as they analysed blood samples from 201 patients with a history of WT for constitutional WT1 mutations, which was not carried out in our cohort of patients [34]. As a result of their work, this underscores the need for a regular screening in patients with GU.

### 4.3. Tumor Volume and Response to Preoperative Chemotherapy

Tumor response to preoperative chemotherapy varies significantly between the different syndromes and depends on the presence of a stromal subtype or WT1 mutation. Thus, we confirm a poor response in patients with DDS [26,27]. These patients have comparably larger initial TVs that can even increase after preoperative chemotherapy (Table 4). A WT1 mutation/deletion path driven propensity for stromal components or even stromal predominance is a likely reason [35,36]. The TV reduction after preoperative chemotherapy is also poor in WAGR patients with WT1 deletion despite of missing stromal type WT (Table 4). In contrast, an excellent response on preoperative chemotherapy is achieved in patients with BWS. As a consequence of a poor TV reduction, a stromal subtype or WT1 aberrations with an underlying syndrome may be possible. This is especially true for patients with bilateral disease.

### 4.4. Bilaterality

The significantly increased frequency of bilaterality in patients with WAGR, GU, DDS is consistent with previous work [9,23,24]. In patients with BWS we found a higher percentage of bilaterality (31.3%) than Breslow et al. (21%) [9]. Consequently, patients with unilateral disease and WAGR, DDS or BWS should always be regarded as predisposed for bilaterality [4] and in cases of bilateral disease these three syndromes need to be kept in mind, if not diagnosed yet.

### 4.5. Metastatic Disease

Metastases are a significantly less frequent event in patients with syndromes as compared to non-syndromic WT. This may be due to an early diagnosis of WT or nephroblastomatosis in patients with syndromes and underscores the importance of a screening program [20].

### 4.6. Histology

The majority of patients with syndromes have intermediate-risk WT. As described earlier, patients with DDS are significantly more often affected by WT with a stromal subtype and never with high risk tumors [24,25]. There is a significant association with blastemal subtype after preoperative chemotherapy in patients with IHH and a trend in BWS due to the frequently IGF2 driven biology. In contrast to the work of Green et al., focal and diffuse anaplasia do not occur as first histology in our data [30]. However, after nephroblastomatosis the development of diffuse anaplasia in case of a secondary WT is a relatively frequent event [37]. We confirm the association between WAGR, BWS and IHH and nephroblastomatosis found by others [9,21,31,33] in contrast to patients with DDS and GU [23]. As nephroblastomatosis is found more often in patients with the above mentioned three syndromes this raises the question whether all patients with nephroblastomatosis need to be examined for CPS if not yet known [33].

### 4.7. Outcome

With the exception of patients with BWS showing a significantly worse EFS and increased risk of relapse, CPS or GU in general have no impact on EFS. Breslow et al. found no difference in WT with BWS neither in OS nor in EFS [9]. However, if nephroblastomatosis is present in our data, EFS tends to be worse, especially if these patients develop a WT as already shown by Furtwängler et al. [37,38]. Therefore, patients with nephroblastomatosis independent of a predisposition syndrome must be followed in regular intervals after the end of treatment for longer periods of time to diagnose a relapse early in order to keep their overall survival as high as for other patients [39].

## 5. Conclusions

Diagnosis of WT at an early age, bilateral tumors or nephroblastomatosis in patients without a known CPS should always raise suspicion of an underlying CPS and genetic testing and counselling should be offered to these patients and families. Screening for WT in patients with a syndrome may stop earlier after the age of 4 to 5 years in patients with DDS and WAGR as also recommended by Hol et al. [8], whereas for the other syndromes this should last up to the age of 7 years and needs to continue in cases of nephroblastomatosis even in CR after the end of first line treatment.

## Figures and Tables

**Figure 1 cancers-13-05016-f001:**
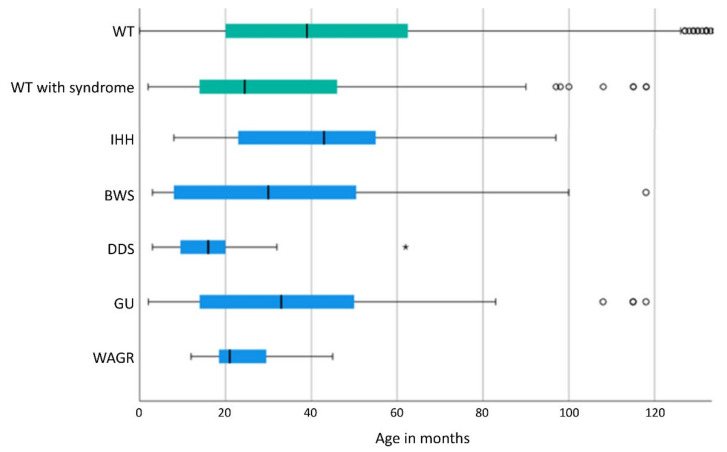
Age of WT/nephroblastomatosis at diagnosis of syndrome-associated WTs displayed as a boxplot (The line in the bar shows the median, the left end of the bar the lower quartile and the right end of the bar the upper quartile, the horizontal line ranges from the minimum to the maximum of data with dots and stars as outliers). DDS and WAGR show a significant lower age at diagnosis compared to all other syndromes (*p* < =0.001). In 3 patients with CPS WT was diagnosed beyond 10 years of age (IHH (173 months), GU (146 months, 325 months)) and in 143 WT without a syndrome beyond 120 months, the last one at 628 months (data not shown). WT: Wilms tumor; GU: Genitourinary malformations; BWS: Beckwith-Wiedemann spectrum; IHH: isolated hemihypertrophy; DDS: Denys-Drash syndrome; WAGR: Wilms tumor, aniridia, genitourinary abnormalities, range of developmental delays.

**Figure 2 cancers-13-05016-f002:**
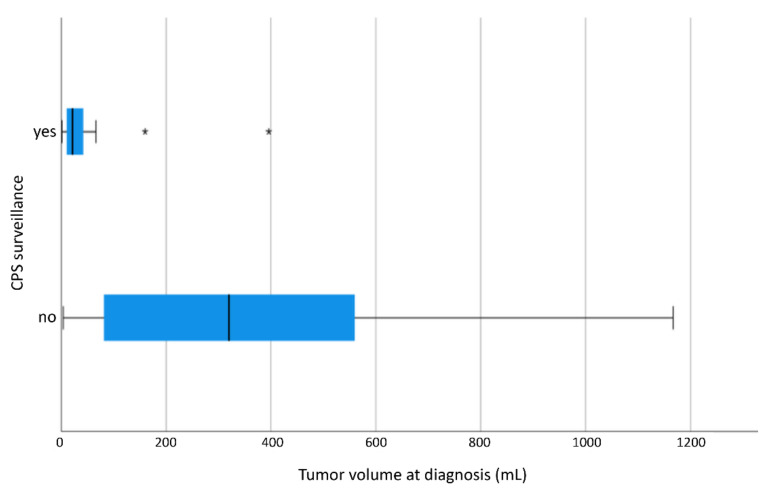
TV at diagnosis of CPS patients as a function of CPS surveillance displayed as a boxplot. TV at diagnosis of 17 patients with CPS surveillance and of 101 patients without CPS surveillance. 2 outliers in patients with CPS surveillance at 160.0 mL and 396.0 mL and 3 outliers in patients without CPS surveillance at 1631.0 mL, 1632.0 mL and 2051.0 mL. CPS patients with CPS surveillance show significantly smaller TV at diagnosis (*p* < 0.001). “*”: outliers; CPS: cancer predisposition syndrome.

**Figure 3 cancers-13-05016-f003:**
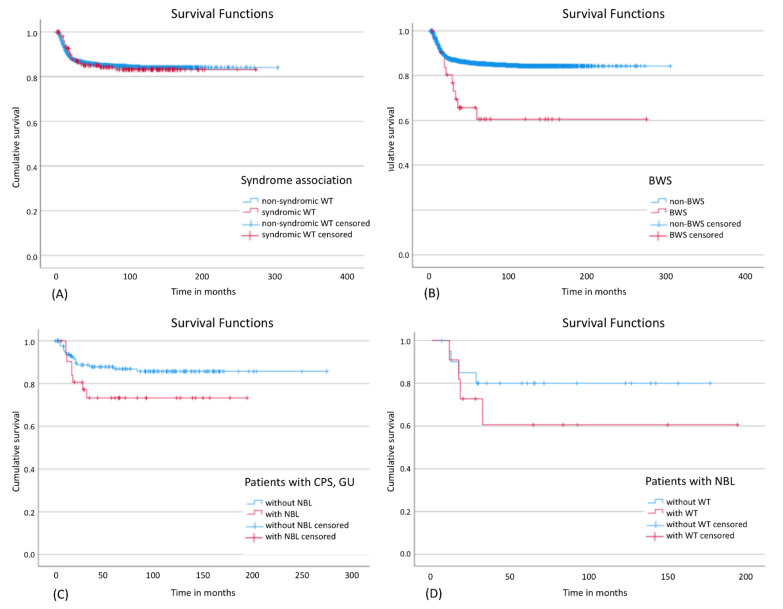
Event-free survival in different subgroups: (**A**) Influence of a syndrome on EFS in patients with WT and/or nephroblastomatosis; Log Rank: *p* = 0.890; (**B**) Influence of BWS on EFS in patients with WT and/or nephroblastomatosis; Log Rank: *p* = 0.002; (**C**) Influence of nephroblastomatosis on EFS in patients with a CPS; Log Rank: *p* = 0.086; (**D**) Influence of WT on EFS in patients with nephroblastomatosis; Log Rank: *p* = 0.315. CPS: cancer predisposition syndrome; BWS: Beckwith-Wiedemann spectrum; NBL: nephroblastomatosis, GU: Genitourinary malformations.

**Table 1 cancers-13-05016-t001:** Frequency of malformations and CPS in WT/nephroblastomatosis (NBL).

	Frequency								
	All Patients with WT and/or NBL			Only Bilateral WT and/or NBL			Only Screened Patients with CPS/Malformation and WT and/or NBL		
All WT	2927	-	100%	253	8.6%	**	29 ***	****	**
WAGR	20	∑ = 171	0.7%	6	2.4%	30.0%	8	27.6%	40.0%
GU	66		2.3%	8	3.2%	12.1%	1	3.4%	1.5%
DDS	24		0.8%	7	2.8%	29.2%	3	10.3%	12.5%
BWS	32		1.1%	10	4.0%	31.3%	10	34.5%	31.3%
IHH	29		1.0%	4	1.6%	13.8%	4	13.8%	13.8%
Other *	27	-	0.9%	7	2.8%	25.9%	3	10.3%	11.1%
All	198	-	6.8%	42	16.6%	21.2%	29	100%	14.6%

* see Appendix A; ** % related to the specific malformations or CPS, *** screening in 29 of 198 patients with malformations or CPS, **** % related to the 29 screened patients with malformations or CPS. WT: Wilms tumor; CPS: cancer predisposition syndrome; NBL: nephroblastomatosis.

**Table 2 cancers-13-05016-t002:** Association between nephroblastomatosis (NBL) and CPS or malformation in the whole cohort of patients and bilateral disease; * chi-square *p* ≤ 0.001. WT: Wilms tumor; CPS: cancer predisposition syndrome; NBL: nephroblastomatosis; GU: Genitourinary malformations; BWS: Beckwith-Wiedemann spectrum; IHH: isolated hemihypertrophy; DDS: Denys-Drash syndrome; WAGR: Wilms tumor, aniridia, genitourinary abnormalities, range of developmental delays.

	Isolated NBL		WT + NBL		WT Only		Total	
Total	73	2.5%	64	2.2%	2790	95.3%	2927	100%
Bilateral disease	31	12.3%	61	24.1%	161	63.6%	253	100%
Patients with CPS or GU	22 *	12.9% *	11	6.4%	138	80.7%	171	100%
Patients without CPS or GU	51	1.9%	53	1.9%	2652	96.2%	2756	100%
WAGR	7 *	35.0% *	2	10.0%	11	55.0%	20	100%
BWS	7 *	21.9% *	3	9.4%	22	68.8%	32	100%
IHH	5 *	17.2% *	0	0.0%	24	82.8%	29	100%
DDS	1	4.2%	2	8.3%	21	87.5%	24	100%
GU	2	3.0%	4	6.1%	60	90.9%	66	100%

**Table 3 cancers-13-05016-t003:** Association between histological subtypes and CPS or malformation; * chi-square: *p* ≤ 0.040. WT: Wilms tumor; GU: Genitourinary malformations; BWS: Beckwith-Wiedemann spectrum; IHH: isolated hemihypertrophy; DDS: Denys-Drash syndrome; WAGR: Wilms tumor, aniridia, genitourinary abnormalities, range of developmental delays.

	Stromal Subtype		Blastemal Subtype after Preoperative Chemotherapy		Other Histological Subtypes	
All WT	270	9.2%	215	7.3%	2442	83.4%
WAGR	0	0.0%	1	5.0%	19	95.0%
GU	4	6.1%	5	7.6%	57	86.3%
DDS	9 *	37.5% *	0	0.0%	15	62.5%
BWS	0	0.0%	3	9.4%	29	90.6%
IHH	0	0.0%	5 *	17.2% *	24	82.8%

**Table 4 cancers-13-05016-t004:** Tumor volume (TV) at diagnosis and volume reduction achieved by preoperative (preop.) chemotherapy, * *t*-test: *p* < 0.001 for lower initial TV in patients with CPS. TV is not available for all patients. Standard deviation (SD); CPS: cancer predisposition syndrome; GU: Genitourinary malformations; BWS: Beckwith-Wiedemann spectrum; IHH: isolated hemihypertrophy; DDS: Denys-Drash syndrome; WAGR: Wilms tumor, aniridia, genitourinary abnormalities, range of developmental delays.

	Mean Tumor Volume (TV) and [SD]						
	at Diagnosis		after Preop. Chemo		Volume Reduction		
Patients without CPS or GU (*N* = 1698)	487.5 mL *	[383.0]	228.0 mL	[279.8]	259.5 mL	[326.7]	53.2%
Patients with CPS or GU (*N* = 91)	349.4 mL *	[381.7]	189.6 mL	[255.7]	159.8 mL	[315.3]	45.7%
WAGR (*N* = 10)	104.9 mL	[179.0]	84.4 mL	[156.8]	20.4 mL	[90.1]	19.4%
GU (*N* = 32)	464.0 mL	[329.3]	254.0 mL	[213.2]	210.0 mL	[281.2]	45.3%
DDS (*N* = 15)	379.3 mL	[256.3]	380.7 mL	[375.0]	−1.4 mL	[194.8]	−0.4%
BWS (*N* = 17)	292.9 mL	[539.1]	38.3 mL	[54.2]	254.5 mL	[514.3]	86.9%
IHH (*N* = 17)	307.7 mL	[416.7]	112.7 mL	[244.7]	195.0 mL	[226.7]	63.4%
Patients with CPS or GU undergoing surveillance (*N* = 11)	62.7 mL	[112.0]	55.4 mL	[142.9]	7.3 mL	[36.9]	11.6%

**Table 5 cancers-13-05016-t005:** Multivariate analysis in patients with a WT and/or nephroblastomatosis and only in patients with a syndrome; *: *p* < 0.05. CPS: cancer predisposition syndrome.

	WT and/or Nephroblastomatosis						CPS Patients					
Factor	Relapse			Death			Relapse			Death		
values	*p*-Value	Hazard Ratio	EFS (%)	*p*-Value	Hazard Ratio	OS (%)	*p*-Value	Hazard Ratio	EFS (%)	*p*-Value	Hazard Ratio	OS (%)
CPS patients	0.594	0.931	83.2	0.139	1.422	88.1						
bilaterality	0.000	1.579 *	73.4	0.030	1.976 *	88.2	0.003	3.013 *	65.4	0.639	1.861	85.6
nephro- blastomatosis	0.005	1.220	72.1	0.074	0.266 *	96.2	0.032	1.264	73.3	0.167	0.225	96.7

**Table 6 cancers-13-05016-t006:** Summary of results in the top 5 syndromes associated with WT. ***** significant results (*p* < 0.05); ** isolated nephroblastomatosis, Standard deviation (SD), Standard error (SE). BWS: Beckwith-Wiedemann spectrum; IHH: isolated hemihypertrophy; DDS: Denys-Drash syndrome; WAGR: Wilms tumor, aniridia, genitourinary abnormalities, range of developmental delays.

	Prevalence(%)	Median Age at Diagnosis, [SD] (Month)		Gender	Characteristic Histology	Bilaterality (%)	Average Volume Reduction by Preoperative Chemotherapy	5y-EFS (%), {SE}		Confirmed by
WAGR	0.7	21.0 *	[9.0]	m < f	NBL **	30.0 *	19.4 %	87.5	**{**0.1**}**	[9,21,22]
GU	2.3	33.0	[47.3]	m > f *	-	12.1	45.3 %	87.6	**{**0.4**}**	[23]
DDS	0.8	16.0 *	[12.2]	m > f	stromal type	29.2 *	−0.4 %	94.7	**{**0.5**}**	[24,25,26,27]
BWS	1.1	30.0	[29.7]	m < f	NBL **	31.3 *	86.9 %	60.6 *	**{**0.1**}**	[28,29]
IHH	1.0	43.0	[34.2]	m < f	NBL **/blastemal type after preop. chemo	13.8	63.4 %	84.6	**{**0.1**}**	[6,15,30,31]

## Data Availability

The data presented in this study are available on request from the corresponding author. The data are not publicly available due to ongoing analysis.

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
