# Peer review of "Characteristics of Nephroblastoma/Nephroblastomatosis in Children with a Clinically Reported Underlying Malformation or Cancer Predisposition Syndrome"

_cancers, 2021, doi:10.3390/cancers13195016_

Round 1

Reviewer 1 Report

The retrospective work is of excellent profile on a pathology of significant
cclinical importance in the field of pediatric oncology with a very large case
history with very detailed data analysis.
Excellent style and use of the English
language.
Detailed and updated bibliographic references.  

Author Response

Reviewer 1:

The retrospective work is of excellent profile on a pathology of significant clinical importance in the field of pediatric oncology with a very large case history with very detailed data analysis. Excellent style and use of the English language. Detailed and updated bibliographic references.  

We thank the reviewer for this review.

Reviewer 2 Report

Thank you for the opportunity to comment on the submitted paper on a retrospective analysis of a huge cohort collected during the consecutive GPOH and SIOP Wilms Tumor studies of a 27 year period. The paper provides very helpful information with regard to the clinical presentation and outcome of WT with a cancer predisposition syndrome (CPS). The most important message, that is potentially relevant for follow up investigations after WT, are that patients with WT in the context of CPS less frequently show metastatic tumors. Nevertheless, outcome may be significantly inferior compared to those without CPS.

Nevertheless, a few point should be addressed critically and considered during revision of the paper:

one of the weaknesses is the long observation period in combination with a retrospective design. During this period the access to and methodological approach to screening for CPS has changed significantly. This may explained why the proportion of CPS patients in this cohort is strikingly lower than in other studies (see introduction). Thus, many CPS patients may have been missed in this analysis and would have been detected with reassessment with modern methods. I think that the authors should discuss this critically.

second, the GU group of CPS associated WT, is poorly defined. Which malformations have been included in this study to fulfill the criteria of GU?

The authors report on bilaterality. However, metachronous tumors have only been reported in the context of BWS (n=3). Is this true? and how has NBL presented in the context of CPS: did the authors see Uni- or bilateral NBL? I would appreciate if this information could be added.

Regarding the discussion of the CPS, I would propose that the authors try to separate clearly between the WT associated CPS (DDS, WAGR) and the imprinting disorders (BWS, IHH), because the biological mechanisms are different

Minor comments:

Many tables transport abundant information, in particular table 1, which is so complicated that it would strongly benefiting by letting some aspects out or cutting the table into two tables

Table 6 is not statistically sound, (median) Age at diagnosis and outcome should be given with SD

in table 6 I would propose to report the tumor volume as percentage instead of absolute volumes, because the starting volumes are so variable. And to be precise: this is the “average” volume reduction, and to be statistically sound, it should be given with mean and SD. 

There are two points where the referencing has not worked (page 6, 1st par, page 10, 2nd par)

I have not checked completely, but I have the impression that not all abbreviations have been explained at first occurrence.

in summary, important paper. Congratulations to this important work. I am sure that after revision, this will be a splendid paper that will draw a lot of attention!

Author Response

Reviewer 2:

Thank you for the opportunity to comment on the submitted paper on a retrospective analysis of a huge cohort collected during the consecutive GPOH and SIOP Wilms Tumor studies of a 27-year period. The paper provides very helpful information with regard to the clinical presentation and outcome of WT with a cancer predisposition syndrome (CPS). The most important message, that is potentially relevant for follow up investigations after WT, are that patients with WT in the context of CPS less frequently show metastatic tumors. Nevertheless, outcome may be significantly inferior compared to those without CPS.

Nevertheless, a few points should be addressed critically and considered during revision of the paper:

One of the weaknesses is the long observation period in combination with a retrospective design. During this period the access to and methodological approach to screening for CPS has changed significantly. This may explain why the proportion of CPS patients in this cohort is strikingly lower than in other studies (see introduction). Thus, many CPS patients may have been missed in this analysis and would have been detected with reassessment with modern methods. I think that the authors should discuss this critically.

Thanks for this comment. For this purpose, we have checked the dates of diagnosis to see if there are more patients diagnosed with CPS during successive time periods. Interestingly there was no increase of the percentage of patients with CPS or GU over time. Up to year 2000 54,9% of CPS or GU were diagnosed and 45.1% in the following years.

Second, the GU group of CPS associated WT, is poorly defined. Which malformations have been included in this study to fulfill the criteria of GU?

Thanks for the comment. This is indeed poorly defined. The problem is that more details were not asked about GU malformations. It was only asked if a GU malformation was diagnosed. This may also result in a difference between GU malformations between boys and girls. Unfortunately, this cannot be answered in more detail due to missing data.

The authors report on bilaterality. However, metachronous tumors have only been reported in the context of BWS (n=3). Is this true? and how has NBL presented in the context of CPS: did the authors see Uni- or bilateral NBL? I would appreciate if this information could be added.

Thanks for these questions. We have checked our database. Indeed, one patient with WAGR developed after 7 years and one with IHH after 10 months from initial diagnosis a WT in the contralateral kidney. The text is updated accordingly in chapter 3.8. Outcome.

Regarding NBL, we added a line in Table 2 that displays the frequency of bilateral disease in patients with isolated NBL, WT with NBL and WT only.

Regarding the discussion of the CPS, I would propose that the authors try to separate clearly between the WT associated CPS (DDS, WAGR) and the imprinting disorders (BWS, IHH), because the biological mechanisms are different

Thanks for the remark. We have added a sentence in the discussion: In this respect, it is important to separate clearly between the WT1 associated CPS (DDS and WAGR) and the imprinting disorders (BWS and IHH) showing differences in WT characteristics, e.g. age at diagnosis or the response to preoperative chemotherapy as shown in our analysis.

Minor comments:

Many tables transport abundant information, in particular table 1, which is so complicated that it would strongly benefiting by letting some aspects out or cutting the table into two tables

This figure is updated to make it better readable. A cutting into two tables is no longer needed.

Table 6 is not statistically sound, (median) Age at diagnosis and outcome should be given with SD

We have updated the Standard Deviation for the age at diagnosis and for outcome, we did use the standard error as this is recommended for Life tables.

In table 6 I would propose to report the tumor volume as percentage instead of absolute volumes, because the starting volumes are so variable. And to be precise: this is the “average” volume reduction, and to be statistically sound, it should be given with mean and SD. 

Thanks for this comment. The table is accordingly updated. In addition, we have now only included those patients in table 6 where a tumor volume was available at diagnosis and after pre-operative chemotherapy. This resulted in the mean tumor volume reduction per patient and not for the cohort of patients. The numbers in the table and the text are updated accordingly.

There are two points where the referencing has not worked (page 6, 1st par, page 10, 2nd par)

I have not checked completely, but I have the impression that not all abbreviations have been explained at first occurrence.

This is done in the paper and accordingly updated.

In summary, important paper. Congratulations to this important work. I am sure that after revision, this will be a splendid paper that will draw a lot of attention!

Thanks for these kind words

Reviewer 3 Report

Dear authors,

Authors provide a concept that screening for WT in patients should be considered in patients without a known CPS. This work demonstrated tremendous work. However, major concerns are that this work lacks many efforts to describe their data in result sections.

Major concerns:

  1. For the introduction, authors should introduce current therapeutics or diagnosis/guidelines for WT. Further, authors may emphasize the urgent issues you want to address from your work in the introduction.
  2. For my opinion, the description of results 3.1 should be improved. Authors should deliver a general idea to readers that what the table 1 is going to show.

For instance, authors should mention how many subjected were being studied and types of WT were going to be demonstrated at the beginning of this paragraph.

  1. In result 3.1, authors mentioned “In 137/2927 (4,6%) patients nephroblastomatosis was diagnosed, either isolated (73; 2,4%) or in conjunction with WT (64; 2,1%).” Are these data included in the table 1? These data confused me in the result 3.1.
  2. For result 3.5 Metastatic disease in patients with CPS or malformations, authors should include a table with data to support their result.
  3. For result 3.6, authors should provide a more detailed description of table 2 and table 3 and include any data you want to stress in this result. This would be a more logical way to guide readers to read your work.
  4. For discussion 4.2, authors should give a reason why your data has a big difference from Dr. Diller et al?
  5. For result 3.7 and table 4, authors should state what general chemotherapies/approaches were used in this study? Did the same syndrome in WT use similar preoperative therapy? It is possible that different drugs or therapies but not CPS per se influence tumor volume?

Minor concerns:

  1. Multiple “(Error! Reference source not found.)” appeared in this article. This issue should be corrected.

At the current state, this manuscript needs major revisions for the content and data description.

Author Response

Reviewer 3:

Authors provide a concept that screening for WT in patients should be considered in patients without a known CPS. This work demonstrated tremendous work. However, major concerns are that this work lacks many efforts to describe their data in result sections.

Major concerns:

  1. For the introduction, authors should introduce current therapeutics or diagnosis/guidelines for WT. Further, authors may emphasize the urgent issues you want to address from your work in the introduction.

Thanks for this comment. The paper is accordingly updated:

According to SIOP clinical studies and trials patients are diagnosed by imaging studies alone and preoperatively treated with AV (actinomycin and vincristine) with localized or with (AV plus doxorubicin for 6 weeks) with metastatic tumors. During the registration process of patients, data on the kind of a cancer predisposition syndromes (CPS) or a malformation are provided by the treating hospital without further specifying malformations. In addition, participation in a surveillance protocol before the diagnosis in case of a CPS is registered in the database.

  1. For my opinion, the description of results 3.1 should be improved. Authors should deliver a general idea to readers that what the table 1 is going to show.

Thanks for this comment. Table 1 shows the frequency of all 2927 patients with all CPS and malformations. The 5 most frequent malformations (171 are clearly given in the table and the other 27 malformations are also shown. The supplemental table 1 lists these rare 27 CPS and malformations. In this respect Table 1 shows the frequency of CPS for malformations for all patients, for only those with bilateral WT +/- nephroblastomatosis and for those with WT and/or nephroblastomatosis that underwent a surveillance program. The table and the paragraph is updated for better understanding.   

For instance, authors should mention how many subjected were being studied and types of WT were going to be demonstrated at the beginning of this paragraph.

This is done. See the comment just before.

  1. In result 3.1, authors mentioned “In 137/2927 (4,6%) patients nephroblastomatosis was diagnosed, either isolated (73; 2,4%) or in conjunction with WT (64; 2,1%).” Are these data included in the table 1? These data confused me in the result 3.1.

Yes, these data are included. Please refer to the updated text of comment 3.1 and table 1.

  1. For result 3.5 Metastatic disease in patients with CPS or malformations, authors should include a table with data to support their result.

Such a table is included as supplemental table 3

  1. For result 3.6, authors should provide a more detailed description of table 2 and table 3 and include any data you want to stress in this result. This would be a more logical way to guide readers to read your work.

The following text is added to 3.6:

According to the SIOP studies and trials histology of WT is classified in low, intermediate and high risk, depending on the availability of diffuse anaplasia, the percentage of necrosis and the percentage of blastema, epithel and stroma in the vital tumor part after preoperative chemotherapy. Blastemal type WT and diffuse anaplasia are high risk tumors. Stromal type is mainly associated with WT1 mutations and is not responding on preoperative chemotherapy despite the fact that patients with stromal type WT have an excellent outcome in case of a localized tumor. The histological risk group together with the local and overall stage defines postoperative treatment. In addition, information about nephrogenic rests or nephroblastomatosis are provided. Mixed type, an intermediate risk tumor,

  1. For discussion 4.2, authors should give a reason why your data has a big difference from Dr. Diller et al?

The following is added to this paragraph:

This difference in median age may be related to a different approach as Dr. Diller et al. analysed blood samples from 201 patients with a history of WT for constitutional WT1 mutations, which was not done in our cohort of patients (34). We included patients with a phenotype of GU in whom we do not know the WT1 status. In addition, as you can see in the updated table 6 the SD of the median age is high with 47.3 months in our cohort.

  1. For result 3.7 and table 4, authors should state what general chemotherapies/approaches were used in this study? Did the same syndrome in WT use similar preoperative therapy? It is possible that different drugs or therapies but not CPS per se influence tumor volume?

Thanks for this remark. Treatment was given in the same way for all patients including those with CPS. Treatment is explained in the introduction as already asked for. It may indeed be possible that other preoperative treatment may result in a better shrinkage of tumor volume. But such data are not available yet.

Minor concerns:

  1. Multiple “(Error! Reference source not found.)” appeared in this article. This issue should be corrected.

This is done.

At the current state, this manuscript needs major revisions for the content and data description.

The paper is updated with the useful comments given by the reviewer. Thanks to the reviewer as this has improved the paper substantially.